# A New *Pseudomonas aeruginosa* Isolate Enhances Its Unusual 1,3-Propanediol Generation from Glycerol in Bioelectrochemical System

Julia Pereira Narcizo [1], Lucca Bonjy Kikuti Mancilio [2,3], Matheus Pedrino [2], María-Eugenia Guazzaroni [2,*], Adalgisa Rodrigues de Andrade [1] and Valeria Reginatto [1]

1   Department of Chemistry, Faculty of Philosophy, Sciences and Letters of Ribeirão Preto (FFCLRP), University of São Paulo (USP), Ribeirão Preto CEP 14040-901, SP, Brazil; juliapnarcizoquimica@usp.br (J.P.N.); ardandra@usp.br (A.R.d.A.); valeriars@ffclrp.usp.br (V.R.)
2   Department of Biology, Faculty of Philosophy, Sciences and Letters of Ribeirão Preto (FFCLRP), University of São Paulo (USP), Ribeirão Preto CEP 14040-901, SP, Brazil; lmancilio@umass.edu (L.B.K.M.); matheus.pedrino.goncalves@usp.br (M.P.)
3   Department of Microbiology, University of Massachusetts, Amherst, MA 01003, USA
*   Correspondence: meguazzaroni@ffclrp.usp.br

**Abstract:** The ability of some bacteria to perform Extracellular Electron Transfer (EET) has been explored in bioelectrochemical systems (BES) to obtain energy or chemicals from pure substances or residual substrates. Here, a new pyoverdine-producing *Pseudomonas aeruginosa* strain was isolated from an MFC biofilm oxidizing glycerol, a by-product of biodiesel production. Strain EL14 was investigated to assess its electrogenic ability and products. In an open circuit system (fermentation system), EL14 was able to consume glycerol and produce 1,3-propanediol, an unusual product from glycerol oxidation in *P. aeruginosa*. The microbial fuel cell (MFC) EL14 reached a current density of 82.4 mA m$^{-2}$ during the first feeding cycle, then dropped sharply as the biofilm fell off. Cyclic voltammetry suggests that electron transfer to the anode occurs indirectly, i.e., through a redox substance, with redox peak at 0.22 V (vs Ag/AgCl), and directly probably by membrane redox proteins, with redox peak at 0.05 V (vs Ag/AgCl). EL14 produced added-value bioproducts, acetic and butyric acids, as well as 1,3 propanediol, in both fermentative and anodic conditions. However, the yield of 1,3-PDO from glycerol was enhanced from 0.57 to 0.89 (mol of 1,3-PDO mol$^{-1}$ of glycerol) under MFC conditions compared to fermentation. This result was unexpected, since successful 1,3-PDO production is not usually associated with *P. aeruginosa* glycerol metabolism. By comparing EL14 genomic sequences related to the 1,3-PDO biosynthesis with *P. aeruginosa* reference strains, we observed that strain EL14 has three copies of the *dhaT* gene (1,3-propanediol dehydrogenase a different arrangement compared to other *Pseudomonas* isolates). Thus, this work functionally characterizes a bacterium never before associated with 1,3-PDO biosynthesis, indicating its potential for converting a by-product of the biodiesel industry into an emerging chemical product.

**Keywords:** *Pseudomonas aeruginosa*; bioelectrochemical systems; fermentation; 1,3-propanediol; microbial fuel cells

## 1. Introduction

Microbial Fuel Cells (MFCs) have emerged as an alternative technology for renewable energy generation. MFCs follow a similar concept to traditional fuel cells, but MFCs utilize the catalytic abilities of microorganisms and can use a variety of substrates, converting the energy stored in chemical bonds into electrical current [1]. In this scenario, glycerol is an attractive fuel to be oxidized in MFCs, as it is a versatile and abundant by-product of the biodiesel industry, corresponding to ca. 10% ($v\ v^{-1}$) of total biodiesel produced [2]. Residual glycerol production for the year 2024 is estimated at 680.000 tonnes [3].

Several microorganisms can metabolize glycerol. The usual glycerol metabolism observed in *Klebsiella*, *Clostridium*, and *Citrobacter* genus regenerates NAD+ from the oxidative pathway coupled to 1,3 propanediol synthesis [4–6]. The reductive biochemical pathway converts glycerol to 3-hydroxypropionaldehyde (3-HPA) catalyzed by glycerol dehydratase, which is then reduced to 1,3-PDO by 1,3-PDO dehydrogenase [4–6].

1,3-PDO is used as a monomer in the formation of polymers and directly in end-use products and, therefore, has a high commercial value [7]. Different 1,3-PDO production processes use petroleum derivatives, glucose, or glycerol [8]. However, most 1,3-PDO is synthesized from acrolein or ethylene oxide, both derived from fossil fuels. In the search to increase the competitiveness of biological 1,3-PDO production [4], efforts have been made to improve microbial efficiency in 1,3-PDO-producing microorganisms by metabolic engineering [9,10]. The use of glycerol as a substrate for obtaining 1,3-PDO is an attractive alternative, as it represents the use of a by-product of the biodiesel industry, contributing to the biorefinery concept of this production chain [3].

In previous work, the anodic compartment of an MFC inoculated with a mixed culture from a mining pond was fed with glycerol [11]. After anode biofilm formation, the microbial community of the biofilm was mostly composed of *Citrobacter*, *Pseudomonas*, and *Klebsiella*, at average relative abundances of 45%, 31%, and 6%, respectively. During the MFC feed cycle, 1,3-PDO was produced in the anodic compartment, which was subsequently oxidized and converted to electricity. Both *Citrobacter* and *Klebsiella* are known to form 1,3-PDO from glycerol, and it was assumed that the role of *Pseudomonas* was restricted to aiding in the external electrons transfer (EET) to the electrode. Indeed, *Pseudomonas* species demonstrated a significant role in providing electron shuttles to the anodic microbial communities in MFCs [11,12]. *Pseudomonas* produce phenazines that act as electron carriers and help other microorganisms perform EET, and also transfer electrons from their own respiratory chain to the electrode [13].

In this work, the electrochemical and fermentative performances of a *Pseudomonas aeruginosa* strain EL14 isolated from an MFC anode fed with glycerol [11] were compared with reference strains. EL14 was used as the sole biocatalyst of an MFC bioanode with glycerol as substrate. The products profile and concentration generated in the anodic compartment of the MFC were compared with a fermentative system (FE). Furthermore, genome sequencing of the isolate allowed the investigation of genes that could contribute to the formation of 1,3-PDO, an unusual product of *P. aeruginosa* glycerol metabolism.

## 2. Results and Discussion

### 2.1. Electrochemical Activity of the Isolate

Initially, the oxidation of glycerol on the anodic chamber of the MFC by the *P. aeruginosa* EL14 strain was tested with different external resistances: 500, 1000, 2200, and 3200 $\Omega$. Most investigations employ an external resistor of the MFC in the range of 500–1000 $\Omega$ (Table 1) [14–16]. Here, only the MFC under 2200 and 3200 $\Omega$ produced detectable current. No current was recorded in the abiotic MFC, confirming the electrogenic capacity of the isolate. The current was monitored over 14 days under 2200 $\Omega$, (Figure 1) and a lag-phase of 6 h generally could be detected. During its growth, *Pseudomonas* spp. produce molecules with redox activity associated with mediated electron transfer, such as pyoverdine (PYOv) and pyocyanin (PYOc). The first cycle of the MFC reached a maximum current at $82.4 \pm 5.7$ mA m$^{-2}$ and maintained it for 16.3 h, followed by a gradual current decay. However, the subsequent feeding glycerol cycles produced lower current outputs compared with the first one, $37.6 \pm 7.8$, and $17.7 \pm 4.2$ mA m$^{-2}$ in cycles 2 and 3, respectively. The decrease in the current after the first feeding cycle has also been observed with other *Pseudomonas* species [16]. This effect can be attributed to biofilm ripeness followed by its detachment [17].

**Table 1.** Maximum current density and experimental parameters for MFC fed with glycerol as substrate and *P. aeruginosa* as biocatalyst.

| Strain | $i_{MAX}$ (mA m$^{-2}$) | CE (%) | Glycerol (mM) * | $R_{EXT}$ (ohms) | Anode | Salt Bridge + Cathode | Reference |
|---|---|---|---|---|---|---|---|
| EW819 | 91.3 | 1.49 | 11 | 1000 | Carbon cloth | PEM + Pt-Carbon cloth (air) | [16] |
| ATCC27853 | 110 | --- | 101 | 500 | Carbon cloth | PEM + Pt-Carbon black (air) | [15] |
| EW603 | 141.2 | 5.16 | 11 | 1000 | Carbon cloth | PEM + Pt-Carbon cloth (air) | [16] |
| ATCC27853 | 153 | --- | 101 | 500 | Carbon cloth | PEM + Pt-Carbon black (air) | [15] |
| ATCC27853 | 399.0 | 46.70 | 271.5 | 1000 | Carbon felt | SB + Carbon felt (ferrocyanide) | [14] |
| ATCC27853 | 418.3 | 48.63 | 271.5 | 1000 | Carbon felt | PEM + Pt-Black on a carbon felt with carbon vulcan powder and Teflon (air) | [14] |
| EL14 | 82.4 | 4.54 | 22 | 2200 | Carbon cloth | PEM + Pt-Carbon cloth (air) | This work |

--- Not reported. * Concentrations given in g L$^{-1}$ were converted to mM.

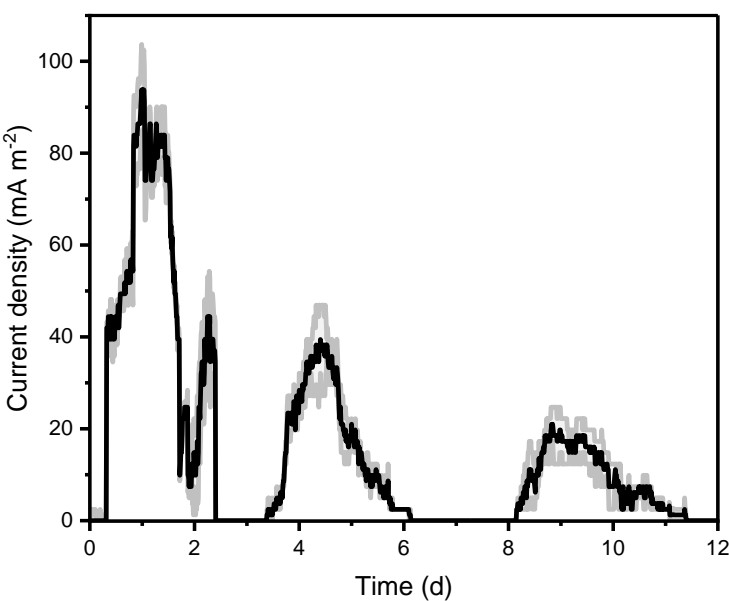

**Figure 1.** Current density (black line) produced in the MFC with an external resistor of 2200 $\Omega$ and glycerol concentration of 22 mM. The shadows represent the standard deviation in the replicates.

Several factors influence the performance of MFCs, including the electron transfer mechanism of the biocatalysts, substrate type and concentration, and reactor design [18]. Nastro et al. (2021) [19] used *P. aeruginosa* ATCC 15,692 in an MFC fed with vegetable residues composed mainly of cellulose, lignin, and starch, and the maximum current obtained in the MFC was 32.9 A m$^{-2}$ per kg of biomass. When glycerol is used as a substrate in MFCs with *P. aeruginosa* as biocatalysts, currents range from 0.63 to 418.3 mA m$^{-2}$ (Table 1). For example, Dantas et al. (2013) [14] reported currents above 300 mA m$^{-2}$, but the glycerol load was 271.5 mM, much higher than that used in our work (22 mM). In addition, in the aforementioned work, the anode and cathode products were continuously recirculated. However, our results are in agreement with MFCs operating under similar

conditions. Zani et al. (2023) [16] reached 91.3 mA m$^{-2}$ by operating a *P. aeruginosa* MFC fed with glycerol at 11 mM. Similarly, Gomes et al. (2011) [15] employed *P. aeruginosa* ATCC27583 and recorded 110 mA m$^{-2}$ (Table 1).

The Coulombic efficiency (CE) furnishes a measurement of the efficiency of charge transfer from the substrate to the anode. A CE around 4.5% is similar to other *P. aeruginosa* strains investigated in our group (Table 1). Nevertheless, improvement in the MFC design and cathode efficiency might enhance the CE.

In order to investigate the electrochemical activity and the possible EET mechanism, cyclic voltammetry of the anodic compartment was performed under two conditions: (A) an abiotic anode in the presence of planktonic cells (Figure 2, red line); and (B) the bioanode of the MFC was transferred to a fresh and sterile medium (Figure 2, blue line). In the first case, an oxidation peak was detected at 0.22 V (vs. Ag/AgCl). Thus, planktonic cells and their mediators act by transferring electrons to the electrode and EET occurs through mediated electron transfer (MET). *P. aeruginosa* is known to produce phenazines that act as redox mediators in bioelectrochemical systems. Within this class of molecules, pyocyanin (PYO) and phenazine-1-carboxamide (PCA) are the well-known phenazine derivatives. Furthermore, pyoverdine (PYOv), a yellowish-green siderophore, which consists of a peptide chain (6–12 amino acids) and a chromophore group, can also exhibit redox properties [20]. PYOv oxidation occurs at around 0.2 to 0.3 V vs. Ag/AgCl, at pH 7 [21], which is very close to that observed in Figure 2 (green line), suggesting that PYOv acted as a redox mediator on the EET of El14. In the case of biofilm alone, the oxidation peak observed is shifted toward lower potentials (0.05 V vs. Ag/AgCl) (Figure 2, blue line). This signal should correspond to the redox species located at the cell membrane, promoting a direct electron transfer (DET) via redox-active proteins such as c-type cytochromes in the outer cellular membrane [22], or even in the extracellular matrix [23].

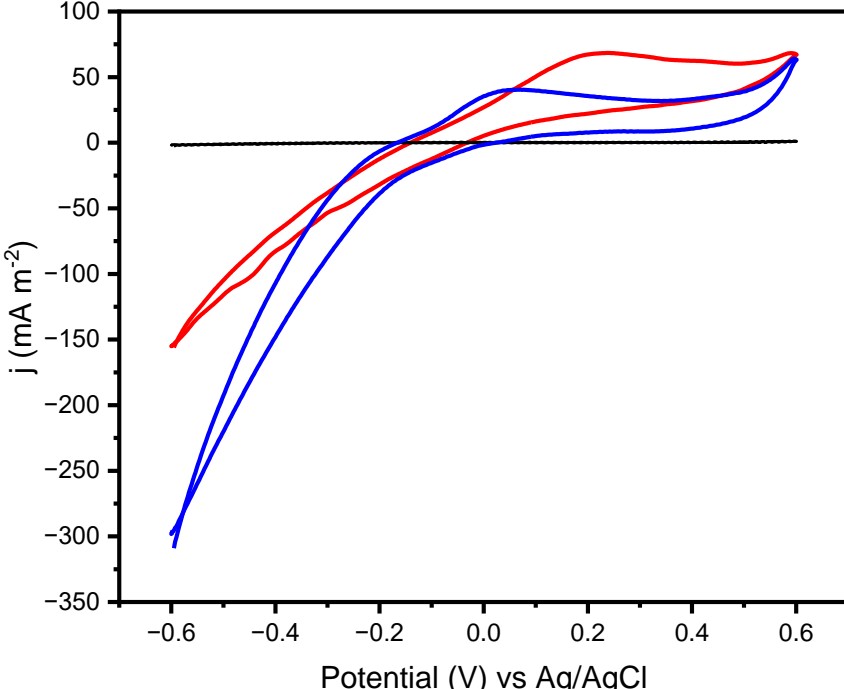

**Figure 2.** Cyclic voltammograms of the anode with planktonic cells (red line) and of the anode with biofilm in fresh and sterile medium (blue line). Abiotic control is represented by the black line. $\nu$ = 2 mV s$^{-1}$, pH 6.9.

### 2.2. Glycerol Oxidation and by-Products Formation during MFC and Fermentative System Operation

The *P. aeruginosa* isolate (EL14) was tested as the biocatalyst in a bioelectrochemical system (MFC) and also in a traditional fermentation system. We used glycerol at 22 mM as substrate, and the products formed were monitored over time in both systems (Figure 3). EL14 produced acetic acid, butyric acid, and surprisingly, 1,3-PDO, in both MFC and fermentation systems.

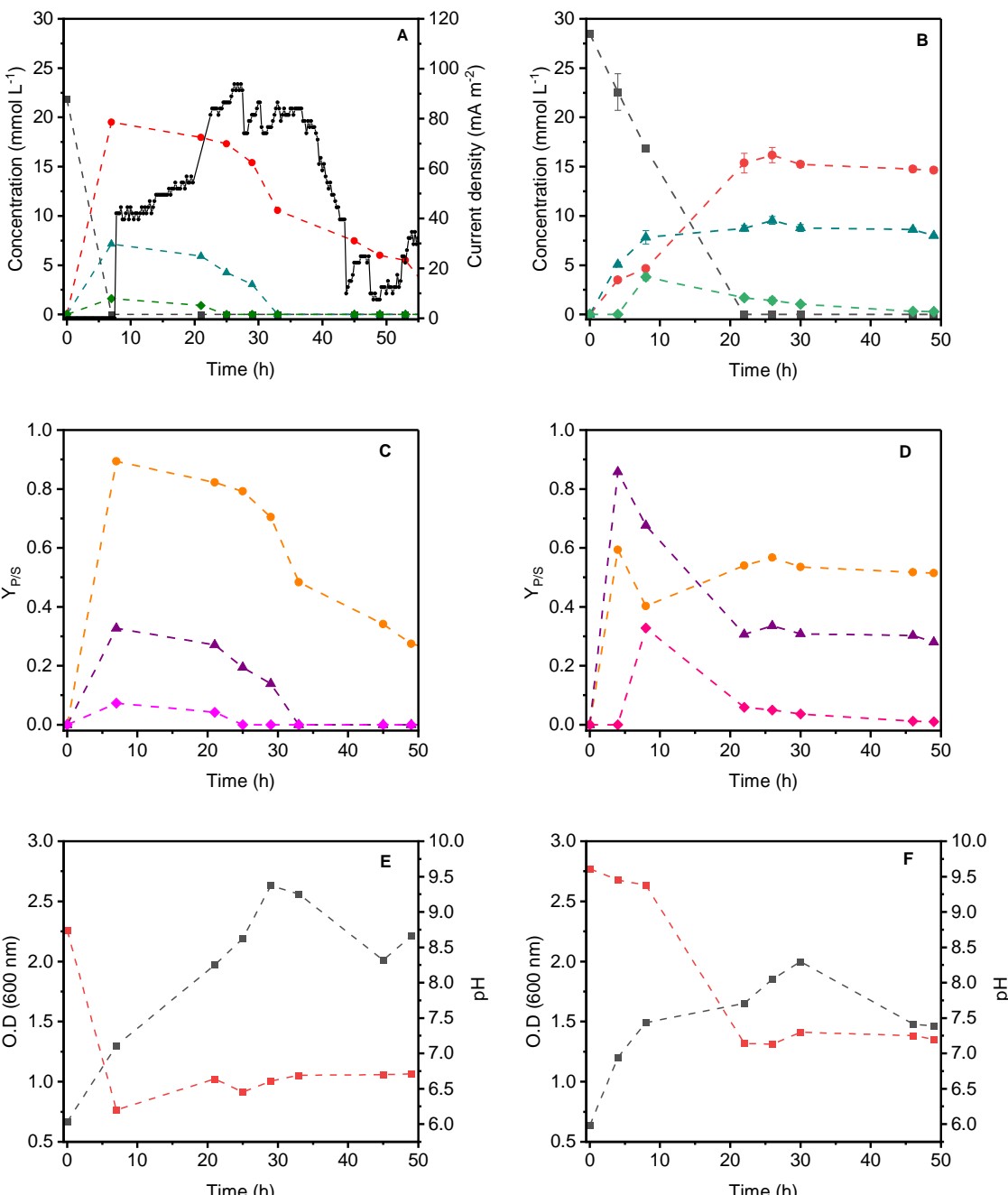

**Figure 3.** Concentration of glycerol (gray line), 1,3-PDO (red line), acetic acid (blue line), and butyric acid (green line), and the current density (black line) produced by the MFC with an external 2200 Ω resistor (**A**) and in the fermentative system over time (**B**). $Y_{P/S}$ conversion factor from glycerol to 1,3-PDO (orange line), acetic acid (purple line), and butyric acid (pink line) during MFC operation (**C**) and in the fermentation (**D**). Optical density at 600 nm (gray line) and pH (red line) in MFC (**E**) and fermentation (**F**).

In the MFC, the by-products were monitored during the first operating cycle with an external resistance of 2200 $\Omega$ (Figure 3A). Glycerol was completely consumed in 7 h of operation with concomitant generation of 1,3-PDO, acetic acid, and butyric acid at 19.5 mM, 7.15 mM, and 1.60 mM, respectively. After that, the by-products were consumed over time, accompanied by an increase in electric current. When organic acids and 1,3-PDO were depleted, electricity could no longer be detected. Thus, the by-products of glycerol metabolism are responsible for the maintenance of the current in the MFC furnishing $82.4 \pm 5.7$ mA m$^{-2}$.

Operating under fermentative conditions, *P. aeruginosa* EL14 produces 1,3-PDO, acetic acid, and butyric acid at concentrations of 16.1, 9.55, and 3.79 mM, respectively (Figure 3B). However, contrary to the MFC system, the concentration of 1,3-PDO and acetic acid remained constant, after 20 h of fermentation. This was not surprising because, in the absence of an electron acceptor (electrode), cells cannot perform EET to gain energy. Therefore, the oxidation of its by-products occurs to a minor extent compared with the MFC. In the fermentative system, only butyric acid is consumed (Figure 3B).

Although the glycerol oxidation by-products detected in the fermentative system and the MFC were similar, the systems differed mainly in their yield of glycerol conversion ($Y_{P/S}$) (Figure 3C,D). The fermentative system showed a higher conversion of glycerol to organic acids (acetic acid 0.86 mol mol$^{-1}$ and butyric acid 0.33 mol mol$^{-1}$) than the MFC (Figure 3C) (acetic acid 0.33 mol mol$^{-1}$ and butyric acid 0.073 mol mol$^{-1}$). However, the yield of 1,3-PDO was higher in the MFC (0.89 mol mol$^{-1}$) compared to the fermentation system (0.57 mol mol$^{-1}$). Interestingly, this could be associated with the high external resistance (2200 $\Omega$) employed in the system, which hindered the flow of electrons from the anode to the cathode. Consequently, electrons from glycerol oxidation can accumulate at the anode/microorganism interface and direct glycerol metabolism to the reducing branch, resulting in 1,3-PDO through alteration of the intracellular oxidation-reduction potential (ORP).

It is known that ORP levels affect many biological functions of cells through gene expression [24,25] and enzyme synthesis, which consequently affect the detection and transduction of signals and, finally, the metabolic profiles [26,27]. In fact, the production of 1,3-PDO was higher in our MFC system than in the fermentative system. Another interesting observation is that in MFC the OD at 600 nm was higher (Figure 3E) than in the fermentation system (Figure 3F), at 2.7 and 2.0, respectively. The higher cell growth in the MFC also relies on an electron surplus for energy generation in the MFC, compared to the fermentation alone.

Moscoviz et al. (2017) [8] reported yields between 0.50 and 0.72 mol for 1,3-PDO/mol of glycerol in the fermentation system. Our *P. aeruginosa* isolate showed promising results, as we obtained maximum yields of 0.57 and 0.89 mol mol$^{-1}$ in the fermentative system and the MFC, respectively. The high conversion of glycerol to 1,3-PDO in the MFC is probably related to the direction of glycerol metabolism toward the reducing branch due to the accumulation of electrons in the working electrode. Our results corroborate the work of Kong et al. (2021) [28], who reported improved productivity of 1,3-PDO (28.3 mM) in a bioelectrochemical system compared to traditional fermentation (11.4 mM), with *Klebsiella pneumoniae* L17 as a biocatalyst.

Another interesting feature is that *P. aeruginosa* has a limited fermentative capacity to produce short-chain organic acids [29,30]. The pyruvate fermentation pathway, which is converted to acetate, lactate, and a small amount of succinate, confers the metabolic ability to survive in oxygen-limited environments [31]. However, 1,3-PDO is rarely associated with the metabolism of *Pseudomonas* species [32,33]. The most common bacteria naturally producing 1,3-PDO include *Citrobacter freundii* [34], *Klebsiella pneumoniae* [35], *Klebsiella oxytoca* [36,37], *Clostridium pasteurianum* [38,39], *Clostridium butyricum* [40], and *Clostridium beijerinckii* [41], which metabolize glycerol to 1,3-PDO through a reductive pathway. In the first step of the reductive pathway, glycerol is dehydrated as a result of the action of the enzyme glycerol dehydratase (GDHt), forming 3-hydroxy propionaldehyde (3-HPA).

Next, the NADH+H$^+$ -dependent 1,3-PDO dehydrogenase (1,3-PDDH) reduces 3-HPA to 1,3-PDO, generating NAD+ again [42,43]. The genomic features of the *P. aeruginosa* isolate EL14 suggest that 1,3-PDO production might occur through alternative routes that do not involve the glycerol dehydratase enzyme, which will be discussed in the next section.

### 2.3. Identification of the Genes Responsible for the Production of 1,3-PDO

Considering that strain EL14 is a promising 1,3-PDO microbial producer, we decided to gain further insight into its potential biochemical features by sequencing its genome. According to whole genome sequencing analysis, strain EL14 clustered with *P. aeruginosa* DSM 50071 with a 93% bootstrap (Figure S1). The highest genomic similarities lay with *P. aeruginosa*, with 95.2% digital DNA–DNA hybridization (dDDH d4) for the EL14 strain.

Next, we wanted to comprehend the genomic context in strain EL14 and its association with 1,3-PDO production. Two key enzymes participate in 1,3-PDO metabolism via a reductive pathway: (i) glycerol dehydratase (*dhaB* gene), which carries out the conversion of glycerol into 3-HPA, and (ii) 1,3-PDO dehydrogenase (*dhaT* gene), which catalyzes the 3-HPA conversion into 1,3-PDO (Figure 4). In this context, we identified three copies of the *dhaT* gene distributed in different *loci* of the EL14 genome. Interestingly, we could not find the *dhaB* gene in the EL14 genome in comparison with other *P. aeruginosa* annotated genomes and even when comparing it with 1,3-PDO microbial producers such as *K. pneumoniae* and *Clostridium* spp. [44]. At the protein level, the identity matrix in alignments showed that all DhaT proteins have between 29.6 and 32.3% of identity, reinforcing the idea that these proteins are highly diverse.

**Figure 4.** Metabolism glycerol in natural 1,3-PDO producers. Pyruvate is converted to different organic compounds depending on the microorganism. 3-HPA, 3-hydroxypropionaldehyde; DHA, dihydroxyacetone; DHAP, dihydroxyacetone phosphate; PEP, phosphoenolpyruvate.

When aligned with DhaT sequences derived from other *P. aeruginosa* isolates that presented in their genomes the 1,3-PDO production pathway, DhaT_3 showed the highest protein percentage of identity (Figure 5). All DhaT proteins from EL14 were also smaller than the other 1,3-PDO and shortened at the N-terminal portion (around 28 aa). In addition, we found that 1,3-PDO dehydrogenases were conserved among strains of *P. aeruginosa* available on NCBI, but these were distinct from *dhaT*_1-3 found in the EL14 strain. Gathering these features, we suggested that the new DhaT-related proteins identified in EL14 might have potential to produce 1,3-PDO in a synergistic way which needs to be explored in future efforts. Moreover, we aim to investigate alternative routes associated with the intermediary 3-HPA [41] that could lead to the production of 1,3-PDO without the involvement of the DhaB protein.

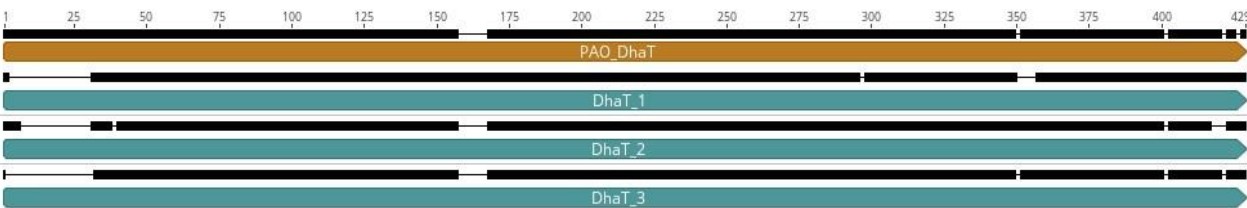

**Figure 5.** Protein sequence alignment of the three 1,3-propanediol dehydrogenases variants (DhaT) identified in the EL14 strain genome and compared with DhaT from *P. aeruginosa* PAO1 (brownish bar). All mismatches among the protein sequences are represented by blank spaces. The upper black bar represents the number of amino acids in the protein sequences.

## 3. Methodology

### 3.1. Bacteria Isolation

Briefly, the isolation of potentially electrogenic bacteria was performed from an electrogenic biofilm formed on the anode of an MFC fed with glycerol [11]. The carbon cloth (9 cm$^2$) anode was washed with 10 mL Lovley and Phillips [45] (the chemicals were purchased from Sigma-Aldrich, Saint Louis, MO, USA) culture medium replacing acetate with 11 mM glycerol and vortexed with glass beads. This suspension was serially diluted to $10^{-2}$ and streaked onto solid media of the same composition. The dishes were placed in anaerobic jars (Anaerogen®, Thermo Fisher Scientific, Waltham, MA, USA) until the appearance of colonies. One colony was picked and named EL14. A CFU resulting from the isolation step was cultured in Lovley liquid medium for DNA extraction.

### 3.2. Cultivation of the Isolate and Culture Medium

The EL14 strain was stored as a liquid stock, containing 15% ($v \, v^{-1}$) glycerol, in Luria Bertani (LB) medium. In the fermentative and bioelectrochemical systems, the culture medium described by Lovley and Philips [45] was used, containing (in mM): 30 NaHCO$_3$, 4.2 Na$_2$HPO$_4$, 5.4 NaH$_2$PO$_4$·H$_2$O, 0.68 CaCl$_2$·2H$_2$O, 1.3 KCl, 28 NH$_4$Cl, 1.7 NaCl, 0.5 MgCl$_2$·6H$_2$O, 0.41 MgSO$_4$·7H$_2$O, 0.025 MnCl$_2$·4H$_2$O, 0.0041 Na$_2$Mo$_4$·2H$_2$O, yeast extract 0.05 g L$^{-1}$, and glycerol at 22 mM, as substrate.

### 3.3. Microbial Fuel Cell Design and Operation

An L-shaped dual-chamber MFC was made of glass, with an anode compartment volume of 35 mL, containing the working electrode, which consisted of a 9 cm$^2$ carbon cloth, fixed with a platinum wire for electric connection. The 16 cm$^2$ cathode (gas-diffusing layer-type carbon cloth, 0.3 mg cm$^{-2}$ PtC 40%) was hot-pressed at 130 °C and 35 kgf cm$^{-2}$ for 180 s with a proton exchanger membrane (Nafion® 117, Sigma-Aldrich, Saint Louis, MO, USA), creating an air-breathing cathode. The external circuit was connected to a 2200 Ω resistor and an Arduino Mega 2560 microcontroller board (Arduino®, Boston, MA, USA) (C++ language) coupled to a PC for the voltage monitoring. Dissolved oxygen was removed from the anodic compartment by bubbling nitrogen.

For the kinetic assays, samples were collected at regular times to determine pH, glycerol, and the by-product concentrations by gas chromatography (GC) (Shimadzu, Kyoto, Japan).

### 3.4. Fermentation System

The fermentative assays were carried out in triplicate in 100 mL flasks containing 50 mL of medium. The culture medium Lovley and Philips [45] was used as described, containing 22 mM of glycerol as a carbon source. To remove dissolved oxygen from the culture medium and inside the flasks, nitrogen gas was bubbled for 10 min prior to autoclaving. After inoculation, flasks were incubated at 37 °C and 150 rpm.

Samples were collected at regular times, from 0 to 25 h after inoculation, to determine pH, glycerol, and the by-product concentrations by gas chromatography (GC).

### 3.5. Substrate and By-Product Analysis

The samples collected from the fermentative and MFC assays were centrifuged at 9000 rpm for 2 min (Universal Centrifuge 320—Hettich®, Tuttlingen, Germany), filtered (PTFE Hydrophilic 0.22 μm—Analítica®, São Paulo, Brazil), and vortexed for 20 s. Glycerol, and the by-products of its metabolism, such as acetic acid, butyric acid, and 1,3-PDO, were quantified using a SHIMADZU Gas Chromatograph with flame ionization detector (FID) and $N_{2(g)}$ as carrier gas [46]. A 250 μL sample was mixed with 750 of ethanol (Ethyl alcohol, pure, 200 proof, HPLC/spectrophotometric, Sigma-Aldrich, Saint Louis, MO, USA) in an Eppendorf flask, and vortexed for 1 min. Next, the diluted sample was centrifuged at 9000 rpm for 4 min and filtered. Analysis was performed on a Stabilwax®-DA column with 30 m length, 0.25 mm diameter, and 0.25 μm bonded film (RESTEK). The method used a temperature of 185 °C for 3 min, followed by heating to 220 °C at 40 °C min$^{-1}$, and 220 °C for 1 min. The carrier gas was $N_2$, used at a linear flow rate of 2.5 mL min$^{-1}$. For the analysis, 2 μL of the sample was injected into the equipment at 300 °C, and a 30:1 split was performed. A flame ionization detector (FID) at 290 °C was also employed.

### 3.6. Conversion Factor ($Y_{P/S}$)

$Y_{P/S}$ corresponds to the conversion factor of the substrate into the product in a given fermentation time interval and is calculated as [47]

$$Y_{P/S} = \frac{P - P_0}{S_0 - S}$$

where $P_0$ and P are the concentrations of the product (M) at times t = 0 and t of the fermentation, respectively. $S_0$ and S are the substrate concentrations (M) at times t = 0 and t of fermentation, respectively. The sum of all conversion factors must not exceed 1. However, the culture medium (Section 2.3) contains not only glycerol as a carbon source, but also yeast extract, which represents a carbon input, increasing conversion rates of organic acids.

### 3.7. Electrochemical Measurements

Cyclic voltammetry was performed between −0.6 and 0.6 V against an Ag/AgCl reference electrode, at a scan rate of 2 mV s$^{-1}$, and data were recorded using AUTOLAB PGSTAT 30 Potentiostat (Metrohm, Riverview, FL, USA) controlled with NOVA 2.1.6 software. The supporting electrolyte was Lovley and Philips medium.

### 3.8. Coulombic Efficiency (CE)

The actual Coulombs transferred per mol of substrate oxidized is determined by integrating the current over time, so that the Coulombic efficiency for an MFC, evaluated over a period of time tb, is calculated as [48]

$$EC = \frac{Cp}{CT_i} \times 100\%$$

where Cp (C) is the total Coulombs calculated by integrating the current over time. $CT_i$ (C) is the theoretical number of Coulombs that can be produced from glycerol,

$$C_{Ti} = \frac{Fb_iS_iv}{M_i}$$

where F is Faraday's constant (98,485 C/mol of electrons), $b_i$ is the number of mol of electrons produced per mol of substrate (14), $S_i$ (g $L^{-1}$) the substrate concentration, $v$ (L) the liquid volume, and $M_i$ the molecular weight of the substrate (92.09 g $mol^{-1}$).

### 3.9. Genome Sequencing and Identification of 1,3-PDO-Associated Genes

Total DNA of *P. aeruginosa* EL14 strain was extracted using the Wizard Genomic DNA Purification Kit (Promega®, Fitchburg, WI, USA) according to the manufacturer's instructions. The concentration of DNA was determined fluorometrically using the Qubit® 3.0 (Qubit® dsDNA Broad Range Assay Kit, Life Technologies, Carlsbad, CA, USA). A DNA library was prepared using the Nextera XT DNA Library Prep Kit (Illumina, San Diego, CA, USA), assessed for quality using the 2100 Bioanalyzer (Agilent Genomics, Santa Clara, CA, USA), and subsequently submitted to sequencing using the Illumina HiSeq (2 × 150 bp) platform (Illumina, San Diego, CA, USA). Paired-end short reads were trimmed and assembled using SPAdes v.3.15.4 [49,50]. The completeness, quality, and contamination were assessed using Type Strain Genome Server (TYGS). Draft genomes were annotated using Prokka v.1.14.5.

The identification of 1,3-PDO-associated genes (*dhaT* and *dhaB*) was first performed by BLASTn. Subsequently, the sequences were extracted, translated to protein, aligned, and compared to the DhaT amino acids sequence of *P. aeruginosa* localized into pathogenicity island PAGI-1 region (GenBank accession number AF241171) using Geneious Prime® v. 2023.1.2 (Biomatters Ltd, Auckland, New Zealand).

### 4. Conclusions

We reported an isolate of *P. aeruginosa* (EL14) used as a biocatalyst in fermentative and MFC systems. In both configurations, EL14 produced value-added products from glycerol: organic acids and 1,3-PDO, a product rarely associated with the metabolism of *Pseudomonas* species. In the bioelectrochemical system, 1,3-PDO yields were higher than those obtained in the fermentative system. This is associated with the high external resistance employed in the system, which made the electron flux from the anode to the cathode difficult. Consequently, the electrons of glycerol oxidation could accumulate at the anode/microorganism interface and drive the glycerol metabolism to the reductive branch that results in 1,3-PDO. The unexpected fermentative profile of the isolate highlights a potential producer of 1.3-PDO. In addition, new DhaT-related proteins identified in the EL14 genome suggest their activity in glycerol and 1,3-PDO metabolism that might act in a synergistic way and in alternative routes without DhaB involvement. EL14 also proved to be a promising biocatalyst for bioelectricity generation; the maximum current obtained is in the same range as other *P. aeruginosa* strains reported.

**Supplementary Materials:** The following supporting information can be downloaded at https: //www.mdpi.com/article/10.3390/catal13071133/s1, Figure S1: Taxonomic affiliation of strain EL14 inferred with TYGS (FastME 2.1.6.1) (Leftor and Gascuel, 2015) from GBDP (Genome Blast Distance Phylogeny) distances calculated from the EL14 strain genome. The branch lengths are scaled in terms of GBDP distance formula d5. The numbers above branches are GBDP pseudo-bootstrap support values >60% from 100 replications, with an average branch support of 95.5%. The tree was rooted at the midpoint.

**Author Contributions:** V.R. and M.-E.G. conceived of the project. J.P.N. conducted the fermentative assays, MFC operation and electrochemical characterization. L.B.K.M. isolated the bacteria. M.P. conducted the nucleic acid extractions and provided the bioinformatics support. A.R.d.A. has helped with the electrochemical results discussions. V.R., M.-E.G. and L.B.K.M. wrote the final draft. All authors have read and agreed to the published version of the manuscript.

**Funding:** The authors would like to thank the São Paulo Research Foundation (FAPESP) (Processes: 2014/50924-4, 2021/07294-6, 2021/010134-7, 2021/01748-5, 2021/01655-7, 2021/09375-3 and 2022/04024-0), and the National Council for Scientific and Technological Development (CNPq) (INCT 465571/2014-0, 150712/2022-7, 167515/2022-5).

**Data Availability Statement:** The *P. aeruginosa* EL14 genome is available at NCBI under GenBank accession number JASMRB000000000.

**Acknowledgments:** The authors also thank CNPq for the Research Productivity Scholarships (Processes: 308914/2019-8, 302750/2020-7, 306601/2022-2).

**Conflicts of Interest:** The authors declare that no conflict of interest could be perceived as prejudicial to the impartiality of the reported research.

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
