# Peer review of "A New Pseudomonas aeruginosa Isolate Enhances Its Unusual 1,3-Propanediol Generation from Glycerol in Bioelectrochemical System"

_catalysts, doi:10.3390/catal13071133_

Round 1

Reviewer 1 Report

This paper reports on a study using Pseudomonas aeruginosa isolate EL14 as a biocatalyst capable of producing 1,3-PDO, acetic acid and butyric acid in an MFC and a fermentation unit using glycerol as a substrate, respectively. The ability to produce 1,3-PDO was greater in the MFC unit. Subsequent sequencing of the EL14 genome identified new evidence for dhaT-related proteins, which may act synergistically and in an alternative pathway without the involvement of dhaB. The content of this paper is relevant to the theme of the journal and a major revision is recommended to adapt the content for publication.

1) Add a diagram of the metabolism of glycerol by MFC and fermentation units to produce 1,3PDO.

2) Adjusted (lines 91 -104) to the last paragraph of the methods section to make the article more logical.

3) At (page 4, line 169), the "i" in the equation symbol for "bi, Si, Mi" is aligned with line 167.

4) The green line in the CV cycle curve of Fig 2 is not clear enough at -0.45V (VS SCE).

5) At (page 7, line 271), "the electrode can function as an electron donor", how can the electrode act as an electron donor?

6) At (page 10, line 344), "the polarization of the bioanode", the author repeatedly mentions electrode polarization, please elaborate on how electrode polarization affects the EET process of the bioanode?

Author Response

Dear Reviewers,

Thank you very much for the suggestions that certainly have improved the quality of our manuscript. We have included a graphical abstract and the references (lines 197-200; 291-294) suggested by the Editors. The answers to the Reviewers' questions are answered below. The changes in the manuscript are marked in red.

Reviewer 2 Report

This manuscript describes the isolation of a new Pseudomonas aeruginosa strain and its application in a glycerol-fed microbial fuel cell (MFC) for the production of 1,3-propanediol (1,3-PDO) as a value-added product. The authors compared the yield of 1,3-PDO in the MFC and fermentation systems, and concluded that it was higher in the MFC, possibly due to the presence of the electrode. The paper is concise and well-written, with sufficient experimental detail. I regret however that the authors didn’t include a comparison with another typical Pseudomonas aeruginosa strain under the same conditions to further support the conclusion of enhanced 1,3-PDO production by Pseudomonas aeruginosa EL14. There are a few other points that require attention:

1.       Line 138: There appears to be a missing "and" between "Glycerol" and "the by-products."

2.       Line 140: It should be “N2(g)”

3.       Line 153: This formula puzzles me. As written, it yields a conversion factor of 1 at all times. For example, assuming that at some point of time 50% of the substrate is converted to product: Y = (0.5 - 0) / (1 - 0.5) = 1. Please, clarify this point.

4.       Figure 3 C, D: I would expect that the sum of all conversion factors at any given time would not exceed 1, which is not the case in the figure. Please provide more clarifications on how the conversion factor was calculated. Additionally, the units used on the graphs (mol mol-1) do not make sense and should be dimensionless (unity).

5.       Line 213: There are no peaks observed at -0.45 V on the voltammograms. What is observed is the start of a reductive wave around -0.2 V, but not a peak. Please make the necessary corrections.

6.       Lines 263-267: The provided explanation is somewhat speculative and should be rewritten. Regardless of the resistance applied, the observed current is positive, indicating that the electrode acts as an electron acceptor and cannot be considered "an external source of electrons for the bacteria." The assumption of a partial negative charge is also invalid without measuring the point of zero charge of the electrode.

Author Response

Dear Reviewers,

Thank you very much for the suggestions that certainly have improved the quality of our manuscript. We have included a graphical abstract and the references (lines 197-200; 291-294) suggested by the Editors. The answers to the Reviewer's questions are answered below. The changes in the manuscript are marked in red.

Round 2

Reviewer 1 Report

The author has answered my questions and is accepted for publication